# Prediction of *PIK3CA* mutations from cancer gene expression data

**Jun Kang** [iD], **Ahwon Lee, Youn Soo Lee**\*

Department of Hospital Pathology, Seoul St. Mary's Hospital, College of Medicine, The Catholic University of Korea, Seoul, South Korea

\* lys9908@catholic.ac.kr

## Abstract

Breast cancers with *PIK3CA* mutations can be treated with *PIK3CA* inhibitors in hormone receptor-positive HER2 negative subtypes. We applied a supervised elastic net penalized logistic regression model to predict *PIK3CA* mutations from gene expression data. This regression approach was applied to predict modeling using the TCGA pan-cancer dataset. Approximately 10,000 cases were available for *PIK3CA* mutation and mRNA expression data. In 10-fold cross-validation, the model with $\lambda = 0.01$ and $\alpha = 1.0$ (ridge regression) showed the best performance, in terms of area under the receiver operating characteristic (AUROC). The final model was developed with selected hyper-parameters using the entire training set. The training set AUROC was 0.93, and the test set AUROC was 0.84. The area under the precision-recall (AUPR) of the training set was 0.66, and the test set AUPR was 0.39. Cancer types were the most important predictors. Both *insulin like growth factor 1 receptor* (*IGF1R*) and the *phosphatase and tensin homolog* (*PTEN*) were the most significant genes in gene expression predictors. Our study suggests that predicting genomic alterations using gene expression data is possible, with good outcomes.

**Data Availability Statement:** All gene expression files are available from the National Cancer Institute (NCI)'s 42 Genomic Data Commons (GDC) website (https://gdc.cancer.gov/about-data/publications/pancanatlas).

## Introduction

Targeted therapy has become a standard treatment for many cancer patients, however the approach requires a test for a specific cancer genomic alteration, to treat patients. Several direct genomic alteration tests have been developed and proven for their clinical utility to treat patients [1, 2].

Machine learning approaches can be applied to detect genomic alterations. Machine learning algorithms can build prediction models from a large number of predictors, such as radiomic features [3], pathology image [4] or gene expression data [5]. Because most direct genomic tests are more specific and sensitive than predictive models, machine learning approaches may have limited roles in clinical practice, however, machine learning approaches are ideal when direct tests are unavailable or fail.

RAS pathway activation predictions have been performed using gene expression data [5]. Authors used data from The Cancer Genome Atlas (TCGA), with a supervised elastic net penalized logistic regression classifier, with stochastic gradient descent. Their model

**Funding:** The authors received no specific funding for this work.

**Competing interests:** The authors have declared that no competing interests exist.

performance was 84% with an area under the receiver operating characteristic (AUROC) curve, and 63% with an area under the precision-recall (AUPR) curve. Importantly, these authors suggested their approach could be applied to other genomic alterations.

Breast cancer having *PIK3CA* mutations can be treated using PIK3CA inhibitors, in hormone receptor-positive HER2 negative subtypes [6]. The *PIK3CA* mutation is the second most common driver mutation after *TP53*, and is most frequently detected in endometrial carcinoma (45%), followed by breast invasive carcinoma (24%), cervical squamous cell carcinoma, endo-cervical adenocarcinoma (20%) and colon adenocarcinoma (16%) [7].

*PIK3CA* encodes the p110$\alpha$ catalytic subunit of phosphatidylinositol 3′-kinase (PI3K). PI3K is a protein kinase that phosphorylates phosphatidylinositol 4,5-biphosphate (PIP$_2$) to generate phosphatidylinositol 3,4,5-triphosphate (PIP$_3$). The phosphatase and tensin homolog (*PTEN*) converts PIP$_2$ to PIP$_3$ in contrast to PI3K. PIP$_3$ is a second messenger that activates protein kinase B (AKT), which is a serine/threonine-specific protein kinase. AKT inhibits apoptosis and promotes cell proliferation [8].

We applied a supervised elastic net penalized logistic regression model to predict *PIK3CA* mutations. We wanted to ascertain whether this prediction model approach could be applied not only to RAS pathway activation, but also to *PIK3CA* mutation predictions. The purpose of this study is to investigate the *PIK3CA* mutation prediction performance of machine learning models.

## Materials and methods

### Dataset

We used the TCGA pan-cancer dataset. TCGA archives the following; exome sequencing, gene expression, DNA methylation, protein expression, and clinical data from > 10,000 cancer samples across 33 common cancer types. The TCGA dataset is publically available. *PIK3CA* mutation data was extracted using cgdsr rpackage [9]. Gene expression data was downloaded from the National Cancer Institute (NCI)'s Genomic Data Commons (GDC) website. This archives data for TCGA (https://gdc.cancer.gov/about-data/publications/pancanatlas). Gene expression in the TCGA pan-cancer dataset is batch-corrected with normalization.

The target variable was *PIK3CA* mutation status. *PIK3CA* status was considered positive when the case had the following *PIK3CA* variants (C420R, E542K, E545A, E545D, E545G, E545K, Q546E, Q546R, H1047L, H1047R, H1047Y) which were the target variants of the Therascreen *PIK3CA* RGQ PCR Kit, Qiagen, Hilden, Germany. This kit was approved as a companion diagnostics test to treat with PIK3CA inhibitor by the United States Food and Drug Administration.

### Modeling process

To narrow down potential predictors, genes with a large median absolute deviation (> third-quartiles) were selected. Thirty three cancer type dummy variables were included in predictor variables. We split three-quarters of the dataset into the training set and one quarter into the test set. Yeo-Johnson transformation was performed to correct skewness. Centering and scaling were also performed. All preprocessing was performed using the recipe r package [10]. Penalized logistic regression was applied to prediction modeling. Ten-fold cross-validation with target variable stratification was performed over the hyper-parameter grid: $\lambda$ {$10^{-5}$, $10^{-4}$, $10^{-3}$, $10^{-2}$, $10^{-1}$, $10^{0}$}, $\alpha$ {0.0, 0.25, 0.5, 0.75, 1.0}. Lambda ($\lambda$) is a penalty scaling parameter and alpha ($\alpha$) is a mixing parameter of penalty function $((1 - \alpha)/2 \parallel \beta \parallel_2^2 + \alpha \parallel \beta \parallel_1)$ [11].

## Assessing model performance

Model performance was evaluated using AUROC and AUPR curve approaches. The AUPR approach is more informative than AUROC for imbalanced datasets [12]. The modeling process and assessing model performance were performed with the tidymodels rpackage [13].

# Results

## Dataset summary

10,845 cases were available for both *PIK3CA* mutation and mRNA expression data. 5,128 out of 20,502 genes were included in the modeling process, after filtering for median absolute deviation, as described in the modeling process method. The prevalence rate for *PIK3CA* mutation was 0.11 in all cases. The *PIK3CA* mutation prevalence rate in each cancer type varied. The median prevalence rate of *PIK3CA* mutation for each cancer type was 0.03 (range 0–0.33) (Fig 1).

## Selecting model and performance estimation

For 10-fold cross-validation, the model with $\lambda = 0.01$ and $\alpha = 1.0$ (ridge regression) showed the best performance in terms of AUROC (S1 Fig). The final model was trained with the selected hyper-parameters with the entire training set. The training set AUROC was 0.93 and the test set AUROC was 0.84. The AUPR of the training set was 0.66 and the test set AUPR was 0.39 (Fig 2A).

## Performance of each cancer type

Because *PIK3CA* mutation prevalence varied across cancer types, the performance of each cancer type was investigated. The AUROC and AUPR were positively correlated between the training sets and test sets in cancer type sub-analysis (Fig 2B). The AUPR was high in cancer types with high *PIK3CA* mutation rates such as colon, breast and uterus cancer types. The AUROC did not correlate with *PIK3CA* mutation rates of each cancer type (Fig 2C).

## Important predictors

The top 30 important predictors are shown (Fig 3). The coefficient is the parameter of the predictor which represents the effect of the predictor on prediction. *Insulin like growth factor 1 Receptor* (*IGF1R*) mRNA expression was the strongest negative predictor, and *PTEN* was the strongest positive predictor. Both *IGF1R* and *PTEN* are key players in the tyrosine kinase pathway [8, 14]. The cancer types were important predictors. Some cancer types including uterine carcinosarcoma (UCS), bladder urothelial carcinoma (BLCA), pancreatic adenocarcinoma (PAAD), lymphoid neoplasm diffuse large B-cell lymphoma (DLBC) were the strongest predictors.

# Discussion

Our model showed good performance in predicting *PIK3CA* mutations in various cancer types. Our data suggested that the supervised elastic net penalized logistic regression model could be applied not only to the RAS activation pathway, but also to other genomic alterations. Both the RAS activation pathway and *PIK3CA* mutations are key, common cancer genomic alterations. Because they exert significant effect on gene expression in cancer cells, prediction from gene expression data can be good. However, the supervised elastic net penalized logistic

## PIK3CA prevalence

**Fig 1. Prevalence rate of *PIK3CA* mutations across cancer types.** Cancer type abbreviations are explained in the S1 Appendix.

regression model cannot be generalized or applied to other genomic alterations which have have a weak effect on gene expression.

Prediction modeling from the TCGA pan-cancer dataset can be limiting in terms of data preprocessing. The gene expression data is processed by between-sample normalization to remove batch effects. If the model has been trained from between-sample normalization, a new sample cannot be exactly processed with normalization which was done on trainset. A model based on gene expression from the TCGA pan-cancer dataset has limitation in terms of data preprocessing. It is necessary to develop a processing method that is independent of a dataset, to apply gene expression data to the prediction model.

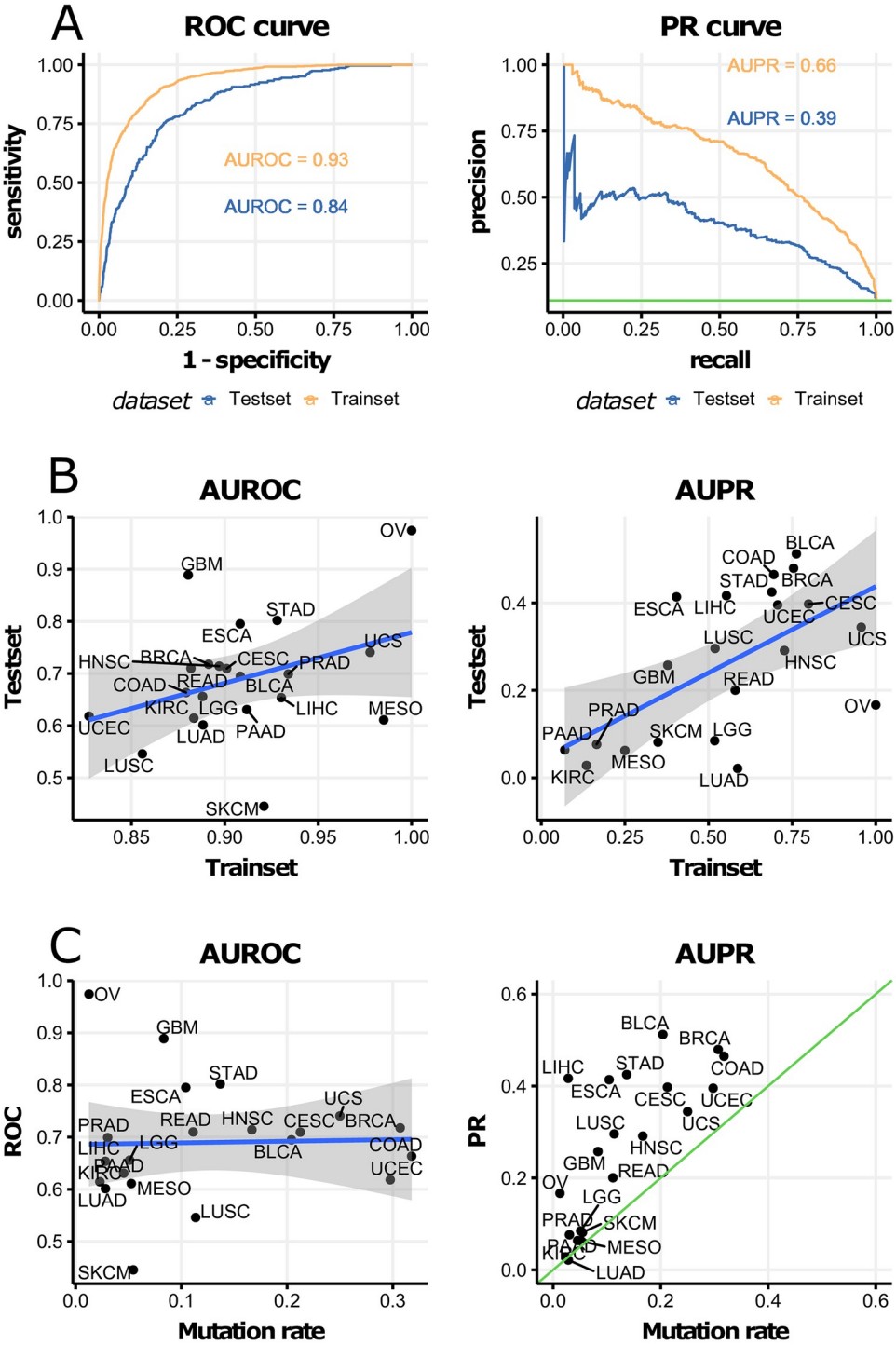

**Fig 2. Summary of modeling results.** (A) Left: receiver operating characteristic (ROC) curve. Right: precision-recall (PR) curve of training set and test set. The horizontal green line is the *PIK3CA* mutation rate (0.11) (B) Correlation between training set and test set of the area under the receiver operating characteristic curve (AUROC), and the area under the precision-recall curve (AUPR) among cancer types. The gray band is the 95% confidence interval. Abbreviations are explained in the S1 Appendix. (C) Correlations between the *PIK3CA* mutation rate of the AUROC, and the AUPR.

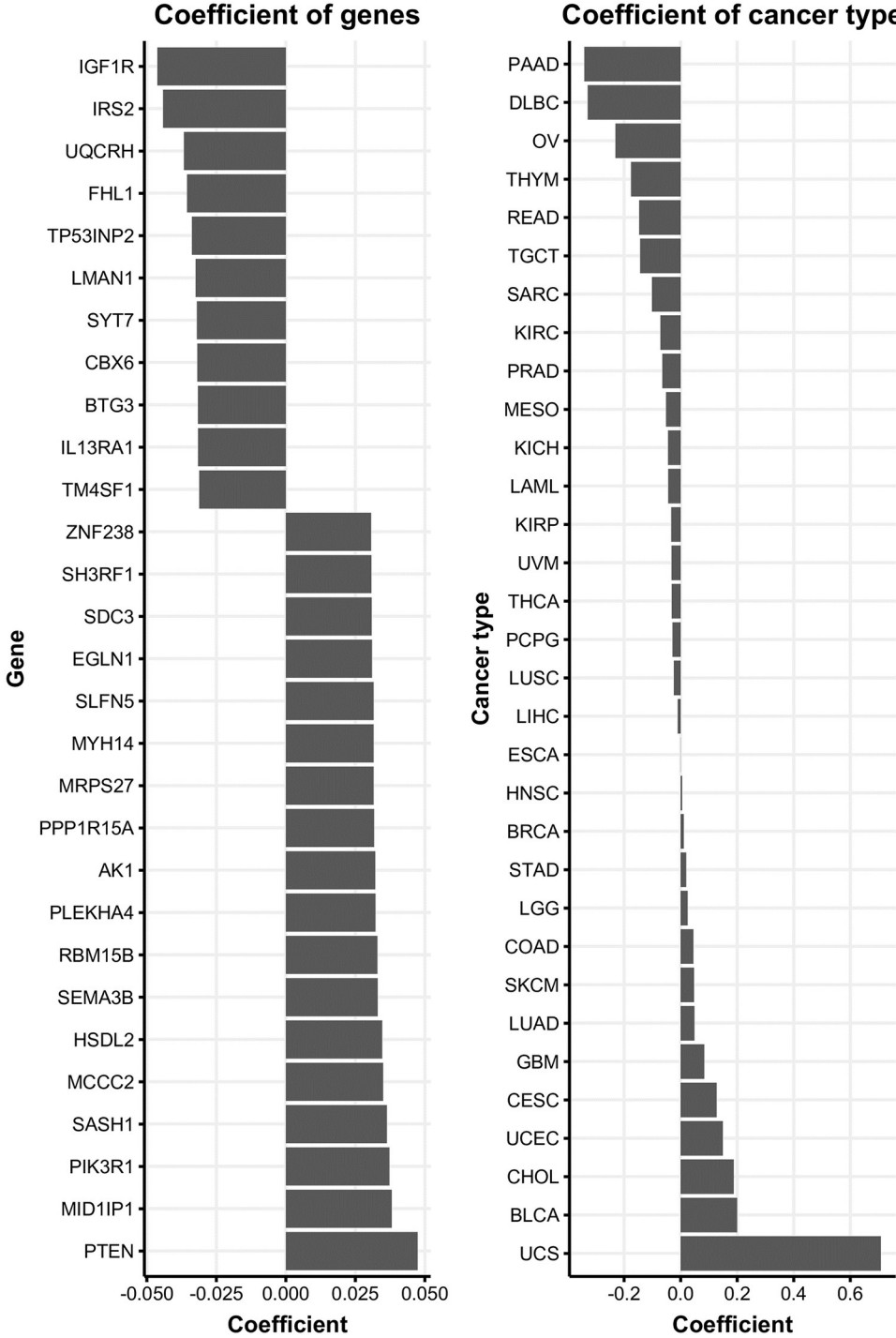

**Fig 3. Coefficient model.** (A) Top 30 high mRNA coefficients. (B) Cancer type coefficients. Cancer types abbreviations are explained in the S1 Appendix.

Our *PIK3CA* prediction model was similar to the RAS activation prediction model in terms of AUROC (0.84). However the AUPR of our model was lower than the RAS activation model (0.39 versus 0.63). The reason for our lower AUPR may be explained by an imbalanced dataset that has the low prevalence rate of *PIK3CA* mutations [5]. The model for RAS activation

trained with cancer types with more than 0.05 prevalence of RAS activation to avoid imbalance classification problem. We included all cancer types in our modeling process. The lower prevalence rate of target variables meant our dataset had a lower AUPR baseline. In the sub-analysis performance of each cancer type, the cancer types with higher *PIK3CA* mutation rates showed better AUPRs.

Our model included cancer types as predictors, and they were stronger predictors than gene expression. The varying prevalence of PIC3CA mutations across cancer types may be a reason for the strong predictive power of cancer types.

Some significant gene expression predictors were closely related to the PTEN-PI3K pathway. *PTEN* and *IGFR1R* were the strongest gene expression predictors, which has negative and positive predictive powers. *IGF1R* is a tyrosine kinase receptor that activates PI3K [14], and *PTEN* is an important regulator of PIP$_3$ by dephosphorylating PIP$_3$ [8].

Several studies have attempted to predict genomic alterations from gene expression data [15, 16]. A study investigated *PIK3CA* mutation predictions using gene-expression signatures which is a sum of the average of the logarithmic gene expression. The model showed good performance AUROC 0.71 in an independent test set [15, 16]. Another study predicted copy number alterations with gene expression, using a multinomial logistic regression model with least absolute shrinkage and selection operator (LASSO) parameters [17]. The prediction of the 1p/19q codeletion was very good, with an AUROC of 0.997, and gene-level predictions were good, with an AUROC of 0.75 [17]. A logistic regression model was used for *MYCN Proto-Oncogene, BHLH Transcription Factor* (*MYCN*) gene amplification in neuroblastoma [18].

The clinical utility of PIK3CA mutation prediction from mRNA expression is unclear because most direct genomic tests are more specific and sensitive than predictive models. Our prediction model is not an application that is immediately applicable to a cancer patient for detection of PIK3CA mutation. It is not known how it will be used, but finding out the mutation prediction performance using gene expression data could play a role in advancing machine learning to be helpful in patient treatment.

Our study suggested that the prediction of genomic alterations using gene expression data was possible, with good performance. However, improved performances are required for clinical tests, and the standardization of generation processing of gene expression data is also needed.

## Supporting information

**S1 Appendix. Abbreviations of cancer type.**
(PDF)

**S1 Fig. Hyperparameter tuning and performance assessment in 10-fold cross-validation resampling.** The x-axis is a penalty scaling parameter: λ {$10^{-5}$, $10^{-4}$,$10^{-3}$,$10^{-2}$,$10^{-1}$, $10^{0}$}, color is mixture hyperparameter of penalty function: $\alpha$ {0.0, 0.25, 0.5, 0.75, 1.0}. y-axis is estimates of area under the receiver operating characteristic (AUROC) using 10-fold cross-validation resampling. (TIF)

## Author Contributions

**Conceptualization:** Jun Kang, Ahwon Lee, Youn Soo Lee.

**Data curation:** Jun Kang.

**Formal analysis:** Jun Kang.

**Methodology:** Jun Kang, Ahwon Lee, Youn Soo Lee.

**Project administration:** Youn Soo Lee.

**Supervision:** Ahwon Lee, Youn Soo Lee.

**Validation:** Jun Kang.

**Visualization:** Jun Kang.

**Writing – original draft:** Jun Kang.

**Writing – review & editing:** Ahwon Lee, Youn Soo Lee.

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
