## [Decision Letter · Decision Letter 0]

26 Aug 2020

PONE-D-20-22669

Prediction of PIK3CA mutations from cancer gene expression data

PLOS ONE

Dear Dr. Lee,

Thank you for submitting your manuscript to PLOS ONE. After careful consideration, we feel that it has merit but does not fully meet PLOS ONE’s publication criteria as it currently stands. Therefore, we invite you to submit a revised version of the manuscript that addresses the points raised during the review process.

Please respond to the comments item-by-item to satisfy the reviewers' concerns. Please carefully edit the MS for English corrections and typos.

We look forward to receiving your revised manuscript.

Kind regards,

Nandini Dey, MS., Ph.D

Academic Editor

PLOS ONE

Journal Requirements:

Reviewers' comments:

Reviewer's Responses to Questions

**Comments to the Author**

1. Is the manuscript technically sound, and do the data support the conclusions?

Reviewer #1: Partly

Reviewer #2: Yes

2. Has the statistical analysis been performed appropriately and rigorously? 

Reviewer #1: I Don't Know

Reviewer #2: Yes

3. Have the authors made all data underlying the findings in their manuscript fully available?

Reviewer #1: Yes

Reviewer #2: Yes

4. Is the manuscript presented in an intelligible fashion and written in standard English?

Reviewer #1: Yes

Reviewer #2: Yes

5. Review Comments to the Author

Reviewer #1: This is a regulated regression analysis to find out the most significant variable(s) and that is kept in the final model. This is a very concise article and lack of detail methodology. It has been known for quite some times that PIK3CA GOF mutation is very common (in fact next to TP53) in solid tumors. It is also known that PIK3CA is very much related to PTEN and IGF1R signaling.

The finding is not new and the rationale for this article is not very clear.

More importantly, this type of article is not suitable for PLOS ONE audience.

Authors may consider to some bio-informatics or bio-statistics journal.

Reviewer #2: The authors present a succinct study on the prediction of PIK3CA mutations from gene expression data. This study applies an elastic net penalized logistic regression classifier to the cancer genome atlas (TCGA) pan-cancer gene expression dataset, a method that was previously established for detecting RAS pathway activation. The methods used and the results presented in the figures appear to be appropriate for the work performed. Both the AUROC and AUPRC demonstrate predictive performance well above baseline. Limitations of the approach used were also appropriately discussed. It may be questionable why PIK3CA mutation prediction from mRNA expression is useful when targeted sequencing panels can assay these mutations directly, but it has been proposed elsewhere that clinical transcriptomics may add important functional or phenotypic information. Overall this work demonstrates that machine learning approaches can predict PIK3CA mutation status from gene expression data with a reasonably good level of performance.

6. PLOS authors have the option to publish the peer review history of their article (what does this mean?). If published, this will include your full peer review and any attached files.

Reviewer #1: No

Reviewer #2: No

---

## [Author Response · Author response to Decision Letter 0]

9 Sep 2020

Reviewer #1: 

Comment 1 

This is a regulated regression analysis to find out the most significant variable(s) and that is kept in the final model. This is a very concise article and lack of detail methodology. 

Response

Thank you for your constructive feedback considering the lack of detailed methodology. TCGA is a widely used public data of cancer genomics. The detail of the TCGA pan-cancer data is described in the reference. For the method of prediction modeling, we tried to follow guidelines for developing and reporting machine learning predictive models in biomedical research. We add a supplementary figure to help understand hyperparameter tuning.

Luo, Wei, Dinh Phung, Truyen Tran, Sunil Gupta, Santu Rana, Chandan Karmakar, Alistair Shilton, et al. “Guidelines for Developing and Reporting Machine Learning Predictive Models in Biomedical Research: A Multidisciplinary View.” Journal of Medical Internet Research 18, no. 12 (2016): e323. https://doi.org/10.2196/jmir.5870.

Comment 2 

It has been known for quite some times that PIK3CA GOF mutation is very common (in fact next to TP53) in solid tumors. It is also known that PIK3CA is very much related to PTEN and IGF1R signaling.

The finding is not new and the rationale for this article is not very clear.

Response

Thank you for your opinion. Our study aims to build the PIK3CA mutation prediction model not to search for important variables. Because the penalized logistic regression model is highly interpretable, we were able to find significant variables like IGF1R and PTEN. But this findings of significant variables is not the primary purpose of this study. The purpose of this study is to investigate the *PIK3CA* mutation prediction performance of machine learning models. The purpose of the study is further described in the manuscript.

Comment 3 

More importantly, this type of article is not suitable for PLOS ONE audience. Authors may consider to some bio-informatics or bio-statistics journal.

Response 

Thank you for your suggestion. Since machine learning modeling is complex and has begun to be widely used relatively recently, the audience may lack an understanding of detailed methods. However, our study used a widely used data set (TCGA) and modeling framework (R tidymodels package). We believe our research will benefit audiences interested in applying machine learning to patient care. We also believe that publishers targeting a broad audience are publishing predictive model studies using machine learning.

Reviewer #2: 

Comment 1 

The authors present a succinct study on the prediction of PIK3CA mutations from gene expression data. This study applies an elastic net penalized logistic regression classifier to the cancer genome atlas (TCGA) pan-cancer gene expression dataset, a method that was previously established for detecting RAS pathway activation. The methods used and the results presented in the figures appear to be appropriate for the work performed. Both the AUROC and AUPRC demonstrate predictive performance well above baseline. Limitations of the approach used were also appropriately discussed.

Response 

Thank you for your opinion. 

Comment 2 

It may be questionable why PIK3CA mutation prediction from mRNA expression is useful when targeted sequencing panels can assay these mutations directly, but it has been proposed elsewhere that clinical transcriptomics may add important functional or phenotypic information. 

Response

As you pointed out, the clinical utility of PIK3CA mutation prediction from mRNA expression is unclear because most direct genomic tests are more specific and sensitive than predictive models. Our prediction model is not an application that is immediately applicable to a cancer patient for the detection of PIK3CA mutation. It is not known how it will be used, but finding out the mutation prediction performance using gene expression data could play a role in advancing machine learning to be helpful in patient treatment. We discussed further the limitations of this study in the manuscript.

Comment 3

Overall this work demonstrates that machine learning approaches can predict PIK3CA mutation status from gene expression data with a reasonably good level of performance.

Response 

Thank you for your opinion.

---

## [Decision Letter · Decision Letter 1]

16 Oct 2020

Prediction of PIK3CA mutations from cancer gene expression data

PONE-D-20-22669R1

Dear Dr. Lee,

We’re pleased to inform you that your manuscript has been judged scientifically suitable for publication and will be formally accepted for publication once it meets all outstanding technical requirements.

Kind regards,

Nandini Dey, MS., Ph.D

Academic Editor

PLOS ONE

Additional Editor Comments (optional):

Reviewers' comments:

Reviewer's Responses to Questions

**Comments to the Author**

1. If the authors have adequately addressed your comments raised in a previous round of review and you feel that this manuscript is now acceptable for publication, you may indicate that here to bypass the “Comments to the Author” section, enter your conflict of interest statement in the “Confidential to Editor” section, and submit your "Accept" recommendation.

Reviewer #1: All comments have been addressed

Reviewer #2: All comments have been addressed

2. Is the manuscript technically sound, and do the data support the conclusions?

Reviewer #1: No

Reviewer #2: Yes

3. Has the statistical analysis been performed appropriately and rigorously? 

Reviewer #1: N/A

Reviewer #2: Yes

4. Have the authors made all data underlying the findings in their manuscript fully available?

Reviewer #1: Yes

Reviewer #2: Yes

5. Is the manuscript presented in an intelligible fashion and written in standard English?

Reviewer #1: Yes

Reviewer #2: Yes

6. Review Comments to the Author

Reviewer #1: As a reviewer I am not satisfied the overall approach of the MS and I am also not satisfied after the revision too.

Reviewer #2: In my original review I questioned the utility of PIK3CA mutation prediction from mRNA data. The authors have added text to the discussion section that has addressed this comment appropriately.

7. PLOS authors have the option to publish the peer review history of their article (what does this mean?). If published, this will include your full peer review and any attached files.

Reviewer #1: No

Reviewer #2: No

---

## [Editor Report · Acceptance letter]

23 Oct 2020

PONE-D-20-22669R1 

Prediction of *PIK3CA* mutations from cancer gene expression data 

Dear Dr. Lee:

I'm pleased to inform you that your manuscript has been deemed suitable for publication in PLOS ONE. Congratulations! Your manuscript is now with our production department. 

Kind regards, 

on behalf of

Dr. Nandini Dey 

Academic Editor

PLOS ONE